# The Molecular Mechanism of the TEAD1 Gene and miR-410-5p Affect Embryonic Skeletal Muscle Development: A miRNA-Mediated ceRNA Network Analysis

**DOI:** 10.3390/cells12060943

**Published:** 2023-03-20

**Authors:** Wenping Hu, Xinyue Wang, Yazhen Bi, Jingjing Bao, Mingyu Shang, Li Zhang

**Affiliations:** Institute of Animal Sciences, Chinese Academy of Agricultural Sciences, Beijing 100193, China

**Keywords:** ceRNA network, skeletal muscle, sheep, *TEAD1* gene, miR-410-5p

## Abstract

Muscle development is a complex biological process involving an intricate network of multiple factor interactions. Through the analysis of transcriptome data and molecular biology confirmation, this study aims to reveal the molecular mechanism underlying sheep embryonic skeletal muscle development. The RNA sequencing of embryos was conducted, and microRNA (miRNA)-mediated competitive endogenous RNA (ceRNA) networks were constructed. qRT-PCR, siRNA knockdown, CCK-8 assay, scratch assay, and dual luciferase assay were used to carry out gene function identification. Through the analysis of the ceRNA networks, three miRNAs (miR-493-3p, miR-3959-3p, and miR-410-5p) and three genes (*TEAD1*, *ZBTB34,* and *POGLUT1*) were identified. The qRT-PCR of the DE-miRNAs and genes in the muscle tissues of sheep showed that the expression levels of the *TEAD1* gene and miR-410-5p were correlated with the growth rate. The knockdown of the *TEAD1* gene by siRNA could significantly inhibit the proliferation of sheep primary embryonic myoblasts, and the expression levels of *SLC1A5, FoxO3, MyoD*, and *Pax7* were significantly downregulated. The targeting relationship between miR-410-5p and the *TEAD1* gene was validated by a dual luciferase assay, and miR-410-5p can significantly downregulate the expression of *TEAD1* in sheep primary embryonic myoblasts. We proved the regulatory relationship between miR-410-5p and the *TEAD1* gene, which was related to the proliferation of sheep embryonic myoblasts. The results provide a reference and molecular basis for understanding the molecular mechanism of embryonic muscle development.

## 1. Introduction

Skeletal muscle development, also known as myogenesis, is a complex multistep process that requires very precise, space- and time-controlled regulation [1]. All trunk and limb skeletal muscles are formed by the dermomyotome and the dermomyotome-derived myotome, while head muscles are developed from the cranial mesoderm [2]. It starts with the specification of the myogenic lineage (i.e., mesodermal progenitors) differentiating into myoblasts [1].

In the early embryonic stage, muscle growth and development determine the number and cross-section of muscle fibers, and this biological process needs to go through four key stages. First is the myoblasts’ formation: the specific differentiation of pluripotent stem cells into myogenic progenitor cells is the basis, while the myogenic progenitor cells in the somites migrate to the limb and differentiate into myoblasts. Second is the formation of nascent myotubes: myoblasts gather to form nascent myotubes through primary fusion. Third is the formation of myotubes: myotubes are formed by the secondary fusion of additional mononucleated myoblasts with nascent myotubes under the control of specific factors. Fourth is the maturation of myofibers: with a series of complex processes, when the proliferation of myofibers reaches a certain number, proliferating stops, myofibers become thick, eventually forming mature myofibers [3]. 

Specific myogenic markers are expressed at different stages in cells of myogenic lineage during myogenesis, such as SIX homeobox 1/4 *(Six1/4*), paired box 3 (*Pax3*), paired box 7 (*Pax7*), myogenic differentiation 1 (*MyoD*), myogenic factor 5 (*Myf5*), myogenic regulatory factor 4 (*MRF4*), myogenin (*MyoG*), myosin heavy chain (*MyHC*), and mouse muscle creatine kinase (*MCK*) [3,4]. The members of the *MyoD* family of myogenic regulatory factors (MRFs), *MyoD*, *Myf5*, *MRF4*, and *MyoG*, are master regulators of myogenic determination and differentiation. Transcription factor networks involving *Pax3* are upstream of *MyoD* family proteins and induce myogenic specification of muscle progenitor cells [2].

Most sheep are monotocous, and the embryo size and pregnancy physiological structure are very similar to humans. Therefore, the pregnant sheep is a good model for the study of placental and fetal physiology [5,6,7,8]. Sheep are also a great model for research on embryonic skeletal muscle development. Muscle development during gestation has a potentially lasting effect on postnatal growth performance and muscle maintenance in full-grown animals. Moreover, the growth rate of prenatal skeletal muscle is significantly higher than that in the postpartum period. Previous studies have shown that sheep muscle development continues during the second half of gestation [9]. The histological characteristics of the ovine fetus have shown that myotube fusion, myofiber transformations, and modulation are the main factors that promote late-stage growth [10]. As such, the analysis of the temporal and spatial characteristics of fiber transformations and a comprehensive molecular understanding of the mechanisms that occur during the late stage are of utmost importance to interpreting the mechanism of muscle development.

Understanding microRNAs (miRNAs) and their regulatory functions are key to the central dogma of genetics [11]. The study of competitive endogenous RNA (ceRNA) networks has led to the identification of new regulatory mechanisms for post-transcriptional regulation, which has facilitated the exploration of the regulation mechanism of genetic information [12,13]. The ceRNA network allows for different types of RNA transcripts (long noncoding RNA, circular RNAs, genes, and pseudogenes) to communicate with each other by competing with the binding of shared miRNA binding sites (MRE) [14]. In this study, we constructed a ceRNA network by integrating long noncoding RNA (lncRNA), miRNA, and mRNA data based on the whole transcriptome profile. Although specific functions of most miRNAs have yet to be elucidated, miRNA-mediated regulation is an indispensable part of the gene regulation mechanism [15,16]. Evidence has shown that miRNAs exert extensive regulatory functions on genes, which are closely related to the regulation of gene expression in various cell processes.

In this study, small RNA sequencing data were combined with data on lncRNA and mRNA to construct an integrated ceRNA network. This network was subsequently used to identify key genes, miRNAs, lncRNAs, transcription factors, and signaling pathways. In addition, we verified the gene function of the De-miRNAs and target genes in sheep primary embryonic myoblasts to reveal the molecular mechanism of embryonic muscle development in sheep. 

## 2. Materials and Methods

### 2.1. Sequencing and Quantitative Samples

Nine pregnant Chinese Merino sheep were slaughtered humanely, three of which were 85 days pregnant, 105 days pregnant, and 135 days pregnant. The longissimus dorsi of the embryo on 85 days (D85), 105 days (D105), and 135 days (D135) was selected with three sample replicates for small-RNA sequencing (D85-1, D85-2, D85-3, D105-1, D105-2, D105-3, D135-1, D135-2, and D135-3).

Quantitative real-time PCR (qRT-PCR) was used to validated the small-RNA sequencing data, and the samples were the original longissimus dorsi of Chinese Merino sheep embryos.

Additionally, to examine the expression of key miRNAs and target genes, the longissimus dorsi from the key developmental stages of Hu sheep, including newborn; weaning; and 3-, 4-, 5-, and 6-month-old Hu sheep, as well as 4- and 6-month-old Gangba sheep, were selected and used for the qRT-PCR. 

### 2.2. Small-RNA Sequencing

Total RNA was extracted using TRIzol reagent (Invitrogen, Pleasanton, CA, USA), following the manufacturer’s protocol. Total RNA quantity and purity were analyzed using a Bioanalyzer 2100 (Agilent, Santa Clara, CA, USA) with an RIN number > 7.0. Approximately 1 µg of total RNA was used to prepare a small RNA sequencing library using TruSeq Small RNA Sample Prep Kits (Illumina, San Diego, CA, USA), according to the manufacturer’s instructions. Then, single-end sequencing (36 bp or 50 bp) was performed on an Illumina Hiseq 2500 at the LC-BIO (Hangzhou, China), according to the manufacturer’s protocol. The miRNAs’ raw data were uploaded to the Gene Expression Omnibus (GEO) database (GSE127287), which also contained lncRNA and mRNA data. 

### 2.3. Differential Expression and Function Enrichment Analysis of miRNAs

The raw reads were processed using the ACGT101-miR program (LC Sciences, Houston, TX, USA) to remove adapter dimers, low-quality sequences, short sequences (<18 nucleotides), N-containing sequences, and repetitive sequences. We compared the preprocessed sequences with the mRNA, RFam, and Repbase databases to obtain valid data and perform a length distribution statistical analysis. The differential expression of miRNAs based on normalized deep-sequencing counts was analyzed using edgeR software (*p* ≤ 0.05 and |log2fold-change| ≥ 1) [17,18]. In addition, functional enrichment analysis of the differentially expressed miRNAs (DE-miRNAs) identified Gene Ontology (GO) terms, and Kyoto Encyclopedia of Genes and Genomes (KEGG) pathways were performed using the online resource DAVID (version 6.8) “https://david.ncifcrf.gov/ (accessed on 20 October 2018)” [19].

### 2.4. Construction of the miRNA–mRNA Networks

The biological role of miRNAs mainly involves negatively regulating the expression of downstream genes [20,21]. Therefore, the downstream target genes of differentially expressed miRNAs were predicted based on the MiRanda (v3.3a), TargetScan “https://www.targetscan.org/vert_80/ (accessed on 20 October 2018)”, and miRNet databases “https://www.mirnet.ca/ (accessed on 20 October 2018)” [22,23]. Finally, the target genes were predicted by the upregulated (downregulated) miRNAs, and the downregulated (upregulated) mRNAs were subjected to analysis to select reliable target genes and construct a miRNA–TG network [24].

### 2.5. Construction of the miRNA–TG–Pathway Network 

The DE-miRNAs were uploaded to the BINGO plugin of Cytoscape (version 3.6.1). The gene annotation file of sheep “http://geneontology.org/ (accessed on 20 October 2018)” was downloaded and imported into BINGO (Cytoscape 3.9.1), with a P threshold for drawing [25]. Combined with the results of the miRNA–mRNA network, the same DE genes were extracted and uploaded to the plugin ClueGo and CluePedia (Cytoscape 3.9.1) to enrich the function [26,27]. The significance threshold was set to 0.05, and the minimum limit of gene clustering was 20.

### 2.6. Construction of the Integral lncRNA–miRNA–mRNA Interaction Networks

In our previous study, Miranda software (v3.3a) was used to identify sites that contributed to the interaction of DE-miRNAs and differentially expressed long noncoding RNAs (DE-lncRNAs) [28]. The DE-miRNAs and their target genes were evaluated using the TargetScan database [29,30]. Python was used to build relationships between DE-lncRNA, DE-miRNAs, and DE-mRNAs. Then, a lncRNA–miRNA–mRNA ceRNA network was visualized using Cytoscape, followed by the extraction of modules representing densely connected nodes [31,32].

### 2.7. qRT-PCR of miRNAs and Target Genes in Longissimus Dorsi

For the small RNA sequencing data validation, U6 was used as the reference primer, and the expression levels of 5 miRNAs were validated in small-RNA sequencing tissues. Meanwhile, U6 and β-actin were used as the reference primer, and the expression of miR-410-5p, miR-493-3p, miR-3959-3p, *TEAD1*, *ZBTB34,* and *POGLUT1* were analyzed by qRT-PCR in the longissimus dorsi of Hu sheep and Gangba sheep after birth. All primers were designed for the selected transcripts from the transcriptome database (Appendix A). 

#### 2.7.1. RT-PCR of the miRNAs and Target Genes

Using cDNA obtained from all of the longissimus dorsi of the Hu sheep and Gangba sheep after birth as templates, a PrimeScript RT reagent Kit with gDNA Eraser (Takara, Dalian, China) and synthesis of the miRNA first-strand cDNA (tail addition method) (Sangon, Beijing, China) were used to reverse transcribe the cDNA and miRNA, respectively, and which could be used for the subsequent experiments.

#### 2.7.2. qRT-PCR of the miRNAs and Target Genes 

Using all of the cDNA as templates, qRT-PCR was conducted, and then the target genes and miRNAs were quantitated on a 7500 Real-time system (Applied Biosystems, Waltham, MA, USA), according to the instructions for the TB Green Premix Ex TaqTMII (Takara, Dalian, China). Each sample was run in triplicate.

#### 2.7.3. Statistical Analysis

The relative expression of the transcripts was calculated using the 2^−ΔΔCt^ method. The data were analyzed with the statistical software SPSS 26.0, and the statistical data were reported as the mean ± standard deviation. GraphPad Prism 8 with the one-way ANOVA statistical method was used to detect differences in the expression between the different development stages tissues, and the Student’s *t*-test was used to compare the differences in the expression among the same tissues. a and b indicate a significant difference (*p* < 0.05); A and B indicate an extremely significant difference (*p* < 0.01).

### 2.8. Knockdown TEAD1 Gene Expression in Sheep Primary Embryonic Myoblasts

According to the design principle of siRNA, three pairs of *TEAD1* siRNAs and negative control siRNA were successfully designed and synthesized (Appendix A). The sheep primary embryonic myoblasts, which were cultured, purified, and characterized from sheep embryos [33], were seeded on 12-well plates. On the second day, the medium was changed to Opti-DMEM (Gibco, Grand Island, NE, USA), three *TEAD1* siRNAs and negative control siRNA (siNC) (30 ng/well) were transfected with Lipofectamine 3000 Agent (Invitrogen, Pleasanton, CA, USA), and the untreated group was used as the control. After 6 h, DMEM with 10% FBS was used to replace the medium; 24 h later, the cells were washed twice with cold PBS, the total RNA was then extracted from cells with TRIZOL (Invitrogen, Pleasanton, CA, USA), and the cDNA samples were obtained according to the instructions for the PrimeScript RT reagent Kit with gDNA Eraser (Takara, RR047A, Dalian, China). Finally, qRT-PCR was used to measure the knockdown efficiency of three *TEAD1* siRNAs. 

The siRNA with the highest efficiency was selected to construct the *TEAD1* gene knockdown model (siTEAD1) in sheep primary embryonic myoblasts, and two methods were used to detect the proliferation of myoblasts after transfection. The first method used the cell count Kit-8 (Sangon, Beijing, China), which added 10% CCK-8 solution to the cell culture medium, incubating at 37 °C for 2 h, and measuring the OD values of siTEAD1, siNC, and the control [34] with a 450 nm microplate reader (Tecan Infinite 200 Pro, Männedorf, Switzerland) at 24 h, 48 h, and 72 h after transfection. 

At the same time, tissue culture dishes (µ–Slides and µ–Dishes 35 mm, low, Ibidi) were used to carry out the scratch assay. When the cell fusion degree was 100%, the µ–Slides were carefully removed and replaced with Opti-MEM (Gibco, Grand Island, NE, USA) for culture. Photographic images at 0 h and 24 h were observed and collected using a microscope. Finally, the number of cell proliferation and healing areas in the different treatment groups were analyzed with ImageJ software [35].

The total RNA of the three transfection groups was used for the quantitative analysis of the growth and development of myoblasts and the expression of related genes. Using Pirmer5 to design the related gene primers (Appendix A), the qRT-PCR method was the same as above.

### 2.9. Dual Luciferase Assay

Online software tools (TargetScan and miRBase) were used to predict the targeting relationship between miR-410-5p and sheep *TEAD1* gene (NCBI: XM-0151010093). The sequence approximately 300 bp upstream and downstream of the miR-410-5p binding site in the *TEAD1* gene was cloned, and enzyme digestion sites were added at both ends. This sequence was linked to the polyclonal site of the psiCHECKTM-2 vector (Promega, Madison, WI, USA) through the double-enzyme digestion of XhoI and NotI, and the wild-type (psiCHECK2-TEAD1 3’UTR WT) and mutant-type (psiCHECK2-TEAD1 3’UTR MT) dual luciferase reporter vector of the *TEAD1* gene were constructed by the InFusion method. The primers are shown in Appendix A.

The sheep primary embryonic myoblasts were seeded on 96-well plates (1 × 10^4^ cells/well). The culture medium was changed to Opti-DMEM the next day, and *TEAD1* dual luciferase reporter vectors (100 ng/well) and miR-410-5p mimics/inhibitor/NC (30 ng/well) were transfected with Lipofectamine 3000 Reagent (Invitrogen, Pleasanton, CA, USA). The test groups were as follows: psiCHECK2-TEAD1 3’UTR WT + miR-410-5p, psiCHECK2-TEAD1 3’UTR MT + miR-410-5p, psiCHECK2-TEAD1 3’UTR WT + miR-410-5p inhibitor, psiCHECK2-TEAD1 3’UTR MT + miR-410-5p inhibitor, psiCHECK2-TEAD1 3′UTR WT + NC, and psiCHECK2-TEAD1 3′UTR MT + NC. After 6 h, DMEM with 10% FBS was used to replace the medium, and the untreated group was used as the control. After 24 h, the cells were washed twice with cold PBS, and then the cells were cultured according to the Dual-Luciferase^®^ Reporter Assay System (Promega, Madison, WI, USA) manual, and the dual luciferase activity of each group was detected and calculated by the hluc fluorescence value/hRluc fluorescence value. 

### 2.10. Validation of the Targeting Relationship between the DE-miRNAs and TEAD1 Gene

The sheep primary embryonic myoblasts were seeded on 12-well plates and replaced with Opti-DMEM the next day. The MiR-493-3p mimics, miR-493-3p inhibitor, miR-3959-3p mimics, miR-3959-3p inhibitor, miR-410-5p mimics, and miR-410-5p inhibitor, as well as the negative control mimics NC and inhibitor NC, were transfected into sheep embryonic primary myoblasts (30 ng/well) using Lipofectamine 3000 Agent (Invitrogen, Pleasanton, CA, USA). After 6 h, the medium was replaced with DMEM containing 10% FBS. After 24 h, the cells were washed twice with cold PBS, and then the cellular RNA was extracted with TRIZOL (Invitrogen, Pleasanton, CA, USA). The cDNA samples were obtained according to the instructions for the PrimeScript RT reagent Kit with gDNA Eraser (Takara, Dalian, China) kit. Finally, qRT-PCR was used to verify the efficiency of the overexpression and inhibition of three DE-miRNAs and the amount of the related gene expression. The primer design of the related genes are shown in Appendix A.

## 3. Results

### 3.1. Characterization of the miRNA Expression Profiles

The average valid rate of the raw data obtained from each sequencing library was approximately 76.7%. The length distribution of the total number of valid data was calculated. Certain RNA sequences, including rRNA, tRNA, snRNA, and snoRNA, were searched against and filtered from the raw reads by aligning the sequences to the RFam and Repbase databases. The length distribution showed that 52.6% of 22 nucleotides in length were clean reads; this indicates that most of the clean reads obtained were miRNA. All of the clean reads were aligned to the sheep genome and mammalian miRbase using the ACGT-miR101 program. A total of 4752 miRNAs were detected, 2275 of which have not previously been reported (Appendix A).

### 3.2. Identification of the DE-miRNAs and Functional Analysis

The DE genes were screened using a threshold of *p* ≤ 0.05. In the expression profile, a total of 505 miRNAs were differentially expressed (Appendix A). The genes were classified into five categories according to the expression characteristics of the profiles: incremental type (107), decreasing type (151), high–low–high type (91), low–high–low type (45), and irregular type (111). Then, a pairwise comparison of the three groups based on filter criteria was performed (D85 vs. D105, D105 vs. D135, and D85 vs. D135). As a result, 63 upregulated miRNAs and 43 downregulated miRNAs were found between D85 and D105, 106 upregulated miRNAs and 123 downregulated miRNAs between D105 and D135, and 172 upregulated miRNAs and 112 downregulated miRNAs between D85 and D135. Overlapping statistics were performed, and 16 DE-miRNAs were found to be overlapped in three groups (Figure 1). Detailed information on these 16 miRNAs was extracted from the corresponding miRNA profiles (Appendix A). Among 16 DE-miRNAs, the sequence of chi-miR-450-5p_R + 2 and bta-miR-450 are the same, and the sequence of chi-miR-365-3p and bta-miR-365-3p are the same. After removing the duplicate miRNAs, 14 miRNAs were obtained for the subsequent network construction.

To better understand the function of DE-miRNAs, the GO terms and KEGG pathways were analyzed. A total of 643 unique biological processes and 73 enriched pathways were identified (*p* ≤ 0.05), including signal transduction processes, such as MAPK, Wnt, regulation of actin cytoskeleton, Ras, focal adhesion, and the ErbB signaling pathway, based on the top 20 GO enrichment analysis results of the miRNAs (Appendix A), including biological processes, cellular components, and molecular functions. These results suggest that spatiotemporal embryo development is related to intracellular protein breakdown and maintenance, energy and material metabolism, and signal transduction. Fourteen miRNAs were analyzed further to better understand their functions. Among the top 20 KEGG pathways (Appendix A), the most highly enriched pathway was nucleotide excision repair. Furthermore, many pathways related to metabolism were also enriched, such as the Wnt signaling pathway, insulin signaling pathway, focal adhesion, and ErbB signaling pathway.

### 3.3. miRNA–mRNA Networks

With 505 DE-miRNAs, over 100,000 target genes were screened in the database. Therefore, we focused on 14 important miRNAs selected for target prediction and screened a total of 1137 target genes. Then, by integrating the DE-mRNAs in the profile, 379 target genes were retained, and 944 miRNA–mRNA interaction pairs related to skeletal muscle fiber development were constructed (Appendix A). Each miRNA regulates multiple mRNAs and vice versa. A total of 253 target genes were obtained from seven downregulated miRNAs, and a total of 126 target genes were obtained from seven upregulated miRNAs. Among them, oar-miR-150 was found to regulate the most genes, regulating 100 genes. The average number of target genes regulated by the upregulated and downregulated miRNAs was over 35, except for bta-miR-450a. 

### 3.4. miRNA–TG–Pathway Network 

Based on the miRNA–mRNA analysis, we performed an analysis of the miRNA–TG–pathway network to determine the biological processes and signaling pathways in which the miRNA target genes are mainly involved. From the BINGO results, the DE-genes were mainly concentrated in 198 biological processes (Appendix A), which mainly cover five aspects (multicellular organismal processes, intracellular signal transduction, reproductive processes, animal organ development, and metabolic processes), of which the muscle tissue development, actin filament-based process, muscle structure development, and muscle cell differentiation processes attracted our attention (Appendix A). From a diagram drawn using ClueGo and CluePedia (Appendix A), five pathways were found to be enriched, namely, cytoskeleton organization, positive regulation of synaptic transmission, regulation of small-GTPase-mediated signal transduction, peptidyl-serine phosphorylation, and regulation of neuron differentiation. Over 47% of genes were involved in the cytoskeleton tissue process. Finally, the DE genes involved in these five pathways were extracted, and miRNAs from these DE genes were used to construct a miRNA–TG–pathway regulatory network. In the miRNA–TG–pathway network (Appendix A), the enriched pathways are the insulin signaling pathway, Wnt signaling pathway, cell cycle, focal adhesion, adherens junction, ubiquitin-mediated proteolysis, ECM–receptor interaction, ErbB signaling pathway, tight junction, and p53 signaling pathway. Among them, the focal adhesion pathway was found to enrich the most differentially expressed target genes. Seven miRNAs enriched in this network were used for an in-depth analysis.

### 3.5. Integral lncRNA–miRNA–mRNA Interaction Networks 

By integrating the miRNA data and the previous lncRNA and mRNA data, we performed a ceRNAs analysis. The DE-lncRNAs were screened to predict their target miRNAs, and data including seven miRNAs selected from the miRNA–TG–pathway network were extracted for the construction of the lncRNA–miRNA–mRNA networks. Among them, 14 DE-lncRNAs were able to adsorb oar-miR-493-3p, bta-miR-450a, oar-miR-23b_R + 6, hsa-miR-410-5p, and oar-miR-3959-3p. However, chi-miR-365-3p and chi-miR-133a-5p were not able to bind to any DE-lncRNAs. Then, 1598 edges in the lncRNA–miRNA–mRNA networks were constructed based on the ceRNA mechanism (Figure 2A). By analyzing its topological characteristics, the top 25 nodes with the highest degree were identified (Figure 2B). Among them, oar-miR-493-3p had the highest frequency in the network, and was related to the genes (*TEAD1*, *MAP1B*, *MCM9*, *BRI3BP*, *MPP7*, *DGKE*, *MAN1A2*, and *ZBTB34*) and lncRNAs (MSTRG.4058, MSTRG.4324, MSTRG.2252, MSTRG.3879, and MSTRG.1470), which are considered to be of great significance in ceRNA networks. Furthermore, oar-miR-3959-3p and hsa-miR-410-5p also regulated multiple genes and lncRNAs simultaneously. As such, oar-miR-493-3p, oar-miR-3959-3p, and hsa-miR-410-5p were selected as important miRNAs. Similarly, according to the degree value of the lncRNAs in the network, MSTRG.3533, MSTRG.4324, and MSTRG.1470 were selected as important candidate lncRNAs. 

### 3.6. Validation of the Small-RNA Sequencing Data 

The expression levels of several miRNAs were detected by quantitative real-time polymerase chain reaction (qRT-PCR) analysis to validate the accuracy of the small-RNA sequencing data. As shown in Figure 3, the results of the qRT-PCR were consistent with the sequencing data.

### 3.7. qRT-PCR of DE-miRNAs and Target Genes in Muscle Tissues of Hu Sheep and Gangba Sheep 

Three key DE-miRNAs (miR-410-5p, miR-493-3p, and miR-3959-3p) and three key target genes (*TEAD1*, *ZBTB34,* and *POGLUT1*) were quantitatively analyzed by qRT-PCR in the muscle samples of Hu sheep and Gangba sheep after birth. The results showed that the expression of miR-410-5p decreased gradually in the four developmental stages of Hu sheep, and the expression of miR-410-5p was significantly different at birth and at 6 months old (*p* < 0.05) (Figure 4A). The expression of miR-493-3p was the lowest at 4 months old, but the difference was not significant (Figure 4B). The expression of miR-3959-3p was not significantly different at the age of birth, weaning, and 4 and 6 months (Figure 4C). The expression of miR-410-5p, miR-493-3p and miR-3959-3p in the muscle tissues of Gangba sheep at the age of 4 and 6 months showed no significant difference separately (Figure 4A–C). There was no significant difference in the expression of these key DE-miRNAs in the muscle tissues at the same age of the Hu sheep and Gangba sheep. However, the expression of miR-410-5p in the Hu sheep was lower than that in the Gangba sheep of the same age (Figure 4A). 

*TEAD1* is a common target gene of miR-410-5p and miR-493-3p. The expression of the *TEAD1* gene had basically no change from birth to the age of 4 months, but it started to increase at the age of 5 months and reached the maximum at the age of 6 months; the expression of the *TEAD1* gene of the Hu sheep at the age of 6 months was extremely significantly higher than that at birth, weaning, and 3 and 4 months (*p* < 0.05) (Figure 4D). The expressions of the *TEAD1* gene in the 4-month-old and 6-month-old muscles of the Gangba sheep had no difference. A comparative analysis of the expression of the *TEAD1* gene in the muscle of the two different sheep breeds showed that compared with the Gangba sheep, the expression of *TEAD1* in the muscle of the 6-month-old Hu sheep was significantly increased (*p* < 0.05), but the difference was not significant at the age of 4 months. The expression of the *POGLUT1* and *ZBTB34* genes in the 6-month-old muscle of the Hu sheep also significantly increased (*p* < 0.05) (Figure 4E,F). 

### 3.8. Knocked-Down TEAD1 Expression Inhibited the Proliferation of Sheep Primary Embryonic Myoblasts 

Three pairs of *TEAD1* siRNAs were constructed and transfected into sheep primary embryonic myoblasts, and it was found that the knockdown efficiency of siRNA1 was better, which could reduce the expression of *TEAD1* gene to approximately 40% of the original, and had a lower miss efficiency (Figure 5A). Moreover, we found that the number of myoblasts in the *TEAD1* knockdown group was significantly less than that in the siNC group and control group (Figure 5A).

CCK8 was used to analyze the cell proliferation of the three groups: siTEAD1, siNC, and control. It was found that the cell proliferation of the siTEAD1 group was lower than that of the siNC and control groups at 24 h, 48 h, and 72 h after transfection (Figure 5B). At 24 h after transfection, there were significant differences between the control/siNC group and siTEAD1 group (*p* < 0.05). At 48 h after transfection, there were significant differences between the control/siNC group and siTEAD1 group (*p* < 0.01). At 72 h after transfection, the difference between the control group and siNC group was significant (*p* < 0.05), and the difference between the control group and siTEAD1 group was highly significant (*p* < 0.01).

The results of the scratch assay showed that the amount of cell proliferation in the siTEAD1 group was significantly less than that in the control and siNC groups at 24 h after transfection, and the wound healing area of the control and siNC groups was also larger than that in the siTEAD1 group, indicating that the cell proliferation rate and wound healing degree in the siTEAD1 group were lower than those in the control and siNC groups (Figure 5C,D). A statistical analysis of the wound healing area of the three groups using ImageJ found that the cell proliferation areas in the control and siNC groups were both 0.14% of the total scratch area. However, the siTEAD1 group was 0.08%, significantly lower than the control and siNC groups (Figure 5D).

When the *TEAD1* gene was knocked down, the expression levels of the *MYF5*, *Pax7*, *MyoD,* and *MyoG* genes, which are markers of myoblast proliferation and differentiation, were all downregulated, and the expression levels of *Pax7* and *MyoD* were downregulated highly significantly (*p* < 0.01). In addition, the expression levels of the *TEAD1* transcription target factor *SLC1A5* and the downstream target gene *FoxO3* were significantly downregulated (*p* < 0.05), while the expression of the downstream genes related to the growth and development of muscle cells, such as *Mrpl21* and *Ndufa6*, had no significant changes in the sheep primary embryonic myoblasts (Figure 5E).

### 3.9. Validation of the Targeting Relationship between miR-410-5p and TEAD1 Gene by Dual Luciferase Assay

The online software tools (TargetScan and miRBase) were used to predict the target relationship between miR-410-5p and the sheep *TEAD1* gene. The prediction results showed that the 3’ UTR region (7843 bp–7889 bp) of the *TEAD1* gene and miR-410-5p had an atypical binding site AAAGTGGT. Based on this binding site, *TEAD1* wild-type and mutant-type dual luciferase vectors were constructed. Through dual luciferase activity detection, it was found that the psiCHECK2-TEAD1 3’UTR WT/MT + miR-410-5p inhibitor/NC transfection groups had no significant changes in luciferase activity. However, in the psiCHECK2-TEAD1 3’UTR WT + miR-410-5p mimics transfection group, a significant decrease in luciferase activity was detected. However, the luciferase activity of the psiCHECK2-TEAD1 3’UTR MT + miR-410-5p mimics transfection group remained unchanged (Figure 6), indicating that miR-410-5p could combine with the 3’UTR region of the *TEAD1* gene, thereby reducing the expression of the *TEAD1* gene.

### 3.10. Validation of the Targeting Relationship between DE-miRNAs and TEAD1 Gene

The mimics and inhibitors of the DE-miRNAs (miR-410-5p, miR-493-3p, and miR-615-3p) were transfected into the sheep primary embryonic myoblasts. It was found that the cell proliferation rate of the group transfected with miRNA-410-5p mimics or miR-493-3p mimics was slower than that of the group transfected with miRNA-410-5p inhibitor or miR-493-3p inhibitor after 24 h, respectively (Figure 7A). The qRT-PCR showed that the expression level of the miRNAs in the groups that added three kinds of mimics was significantly higher than that in the inhibitor groups and control groups (*p* < 0.01) (Figure 7B). However, the expression of the *TEAD1* gene was significantly decreased only in the miRNA-410-5p mimics group (*p* < 0.01). This shows that miR-410-5p and the *TEAD1* gene did have a targeting relationship, which can downregulate the expression of the *TEAD1* gene. Although miR-493-3p and the *TEAD1* gene were also predicted to have a targeting relationship, the results showed that miR-493-3p could not downregulate the expression of the *TEAD1* gene. In addition, compared with the miR-410-5p inhibitor group, the expression levels of *Pax7* (*p* < 0.01), *MyoD* (*p* < 0.05), *SLC1A5* (*p* < 0.01), and *FoxO3* (*p* < 0.01) were also significantly decreased in the miR-410-5p mimics group, this result was consistent with the siRNA knockdown experiment. It can also be found that the expression levels of *SLC1A5* (*p* < 0.05) and *FoxO3* (*p* < 0.05) also significantly decreased in the miR-493-3p mimics group compared to the miR-493-3p inhibitor group; this suggests that although miR-493-3p cannot downregulate the expression of *TEAD1*, it may affect the expression of genes in this pathway and cell proliferation by regulating the expression of other genes (Figure 8).

## 4. Discussion

From the data analysis results, in the D85 vs. D135 group, there were a large number of DE-miRNAs and DE-genes. The gene expression patterns of D85 and D135 were very different, as reported in previous studies [36]. Several genes directly or indirectly regulated muscle development and were continuously expressed at different stages. We speculated that the embryo maintained a small range of cell proliferation, differentiation, and migration during the late stage of pregnancy. From a previous WGCNA analysis between the lncRNAs and mRNAs by our group [28], the DE-lncRNAs and DE-mRNAs that targeted a binding relationship with miRNA were predicted through the database. In the present study, we predicted the target relationships between lncRNA–miRNA and mRNA–miRNA based on the DE-lncRNAs, DE-miRNAs, and DE-mRNAs, and we constructed a lncRNA–miRNA–mRNA network based on the whole transcriptome profile. 

Through the construction of multiple networks, the genes and miRNAs considered to be key factors were identified. MiRNAs (miR-493-3p, miR-3959-3p, and miR-410-5p) and lncRNAs (MSTRG.3533, MSTRG.4324, and MSTRG.1470) identified in the ceRNA network were enriched in the energy metabolism, muscle contraction, and oxidative phosphorylation pathways. All pathway analysis results indicated that the metabolic and oxidative phosphorylation pathways are significantly related to muscle development [37]. Heeley et al. showed that high levels of oxidative phosphorylation existed during the rapid development of mammalian skeletal muscle and the formation of myofibrils [38]. Therefore, we predicted that metabolism not only provides energy throughout embryonic skeletal muscle development but may also fine-tune embryonic muscle developmental patterns. The Wnt signaling pathway is a potential downstream target of myostatin, which can promote the growth and hypertrophy of postpartum skeletal muscle [39]. DE-miRNAs and DE-lncRNAs affect insulin by regulating target genes and ultimately participate in the regulation of skeletal muscle cells into adipocytes [40]. These noncoding RNAs (ncRNAs) play important regulatory roles in the induction of adipogenesis in skeletal muscle, adipose tissue, and myoblasts. 

In this study, through the analysis of ceRNA networks, three genes (*TEAD1*, *ZBTB34,* and *POGLUT1*) were identified. *POGLUT1* is associated with muscle disorders. Muscle satellite cells are activated after muscle damage. Studies have shown that mutations in *POGLUT1* inhibit the repair and regeneration of skeletal muscle injury. *ZBTB34* is a new member of the BTB/POZ protein family, the function of which is not yet clear. It can be used as an important transcriptional regulatory protein in the cell growth process by participating in the regulation of the transcriptional activities of certain downstream genes [41]. 

TEADs (transcriptional enhanced associate domain transcription factors) are also called TEAD transcription factors, TEAD1 encodes a ubiquitous transcriptional enhancer factor that is a member of the TEADs family [42]. This protein directs the transactivation of a wide variety of genes and, in placental cells, also acts as a transcriptional repressor. Mutations in this gene cause Sveinsson’s chorioretinal atrophy. TEADs combine with their coactivators to form different complexes that will regulate gene expression. These genes are critical to embryonic development and organ formation (heart and muscle), and they participate in cell death and proliferation [43]. In the Hippo pathway, TEADs form a complex with transcriptional coactivators Yap (yes-associated protein, encoded by the gene Yap1), Taz (transcription coactivator with PDZ-binding motif, encoded by the Wwtr1 gene), and Vglls (vestigial-like proteins) [44]. Together Yap, Taz, and TEADs are the nexus of the Hippo signal transduction network. In addition, the Hippo kinase cascade, which consists of the kinases Mst1 (Stk4), Mst2 (Stk3), Lats1, and Lats2 are also at the center of the Hippo signal transduction network [45] (Figure 8), regulates cell fate and proliferation and apoptosis in organ development [46].

TEADs bind to CATTCC/GGAATG (MCAT or GTIIC motifs) to regulate gene expression, and they are frequently located close to the promoters of cardiac and skeletal muscle genes [44,47], as well as through binding enhancers [48,49]. The transcriptional coactivators Yap and Taz, which bind to and activate Teads, are phosphorylated by the Hippo pathway, which regulates the activity of Teads [44]. Yet, Tead1 binds muscle genes that are suppressed in rhabdomyosarcoma [50], which is consistent with the finding that Yap and Taz can also repress gene expression [51]. Taz and Yap have overlapping functions in promoting myoblast proliferation, but Taz then switches to enhance myogenic differentiation [45]. Watt et al. found that Yap is a regulator of C2C12 myogenesis, and the phosphorylation of Yap is necessary for C2C12 myoblasts to develop into myotubes [52]. Judson et al. identified Yap as a regulator of muscle satellite cell fate decisions. Yap expression rises during satellite cell activation and stays high until after the differentiation versus self-renewal decision is made. Yap’s constitutive expression maintains satellite cells, Pax7^+^ and MyoD^+^, and satellite cell-derived myoblasts, which promote proliferation but block differentiation. That Yap binds to TEADs was confirmed in myoblasts [53].Yap is also a critical regulator of skeletal muscle fiber size. Watt et al. found that protein synthesis and basal skeletal muscle mass are positively regulated by YAP. Mechanistically, they showed that Yap regulates muscle mass via interaction with TEADs [46].

Mechanically induced skeletal muscle growth is regulated by mTORC1, and Yap can regulate mTORC1. In skeletal muscle, mechanical overload promotes the upregulation of Yap expression, and the overexpression of Yap induces hypertrophy of skeletal muscle via an mTORC1-independent mechanism [54]. The mammalian target of rapamycin (mTOR) kinase signaling can be affected by hippo signaling. Yap regulates miR-29 production, which promotes the degradation of the phosphatase Pten. Pten blocks Akt, and Akt subsequently indirectly activates mTOR. The overexpression of Yap in the skin of mice or knockdown of Lats1 and Lats2 in cells increases mTOR activity [55]. Mst1/2 can interact with Akt1 to form complexes, and when Mst1/2 is knocked down, the activating phosphorylation of Akt1 and Akt substrates was reduced [56]. Moreover, Mst1 is phosphorylated by Akt, which inhibits Mst1 activity. Akt also blocks FoxO3 phosphorylation and nuclear translocation [57]. The molecular relationship between Hippo and mTOR signaling suggests that cell proliferation and protein synthesis could be coordinated through this crosstalk, since Yap and Taz stimulate cell proliferation and because mTOR signaling is necessary for protein synthesis during cell proliferation [44] (Figure 8). 

*TEAD1* can specifically regulate the expression of cardiac troponin T, β-myosin heavy chain, smooth muscle α-actin, and skeletal muscle α-actin in mammalian and avian skeletal muscles [58,59]. In research on the growth and development of smooth muscle, the expression of *TEAD1* was significantly regulated in smooth muscle cells and negatively related to the expression of smooth-muscle-specific genes. The *TEAD1* gene can compete with cardiac myocytes to bind to the serum response factor (SRF), destroying the interaction between cardiac myocytes and SRF, thus weakening the expression of smooth-muscle-specific genes [60]. In addition, TEADs also play a crucial role in embryonic development and striated muscle gene expression, and the overexpression of *TEAD1* leads to the accidental activation of *GSK-3α/β* and a decrease in nuclear β-catenin and NFATc1/c3 protein, and then regulates the expression of genes related to slow muscle [61]. At the same time, TEAD1 is critical to the normal development of the heart, and it is related to the expression of heart-specific genes and the hypertrophic response of primary cardiomyocytes to hormone and mechanical stimulation. The overexpression of the *TEAD1* gene can induce the characteristics of cardiac remodeling related to cardiomyopathy and heart failure [62]. In addition, studies have shown that *TEAD1* plays a vital role in myoblast growth, skeletal muscle development, muscle fiber hypertrophy, muscle regeneration, and myocardial development [63,64]. In this study, knocked-down *TEAD1* expression by siRNA inhibited the proliferation of sheep primary embryonic myoblasts. The CCK8 experiments and scratch assay reached the same conclusion. Some studies have shown that the double knockout of the *TEAD1* and *TEAD2* genes leads to decreasing embryonic cell proliferation and increased apoptosis, and *TEAD1*, as the main coactivator of YAP, regulated cell proliferation and survival during mouse development [65]. Here, after the knockdown of the *TEAD1* gene by siRNA or miR-410-5p, the expression levels of the *SLC1A5* and *FoxO3* genes were all significantly downregulated. Relevant studies found that, as the main downstream nuclear effector of Hippo signal transduction, *SLC1A5* was a new *TEAD1* transcription target [59]. *TEAD1* regulates the functions of its downstream target genes Mrpl21, Ndufa6, and Ccne1 in myoblast growth, skeletal muscle development, muscle fiber hypertrophy, and muscle regeneration [63]. *TEAD1* can regulate the expression of *FoxO3*a in skeletal muscle through calcineurin/MEF2/NFAT and IGF-1/PI3K/AKT, thus affecting the formation and transformation of skeletal muscle fiber types [66]. Therefore, during embryonic skeletal muscle development, *TEAD1* may be involved in the regulation of muscle fiber differentiation, with similar results in some studies on poultry, pigs, and mice [67,68]. 

In this study, Gangba sheep are a local resource in Tibet that adapt to a high-altitude environment well. Gangba sheep grow slowly, and the weight at the age of 24 and 36 months are 25 and 35 kg, respectively. Hu sheep are a local breed in the Tai Lake basin of China, which grow rapidly in the early stage, especially at the age of 4 to 6 months. A ram (not wethers) at the age of 6 months can reach 45 kg or even more. The expression of the *TEAD1* gene in 6-month-old Hu sheep was extremely significantly higher than at birth, weaning, and 4-month-old Hu sheep (*p* < 0.01), and the expression of the *TEAD1* gene in the muscle of the two different sheep breeds showed that compared with Gangba sheep, the expression of *TEAD1* in the muscle of 6-month-old Hu sheep significantly increased (*p* < 0.05). The expression of miR-410-5p was significantly lower at 6 months old than at birth in the Hu sheep (*p* < 0.05), and the expression of miR-410-5p in the Hu sheep was lower than that in the Gangba sheep of the age. From the prediction results, *TEAD1* is the target gene of miR-410-5p. This shows that the expression level of the *TEAD1* gene was positively correlated with the growth of sheep, while the expression of miR-410-5p was negative. Here, it was found that miR-410-5p and the *TEAD1* gene did have a targeting relationship, which can downregulated the expression of the *TEAD1* gene in sheep primary embryonic myoblasts. Xia et al. found that miR-410-5p promoted the development of diabetic cardiomyopathy, miR-410-5p was increased in the myocardial tissue of a diabetes mellitus rat model, cell apoptosis was reduced by the inhibition of miR-410-5p, and knockdown of miR-410-5p improved myocardial tissue structure and diabetes-induced cardiac function [69]. Rats on a high-fat diet exhibited LV dysfunction and heart fibrosis. In rats fed a normal diet, miR-410-5p overexpression caused cardiac fibrosis, whereas in rats fed a high-fat diet, miR-410-5p inhibition decreased cardiac fibrosis [70]. These results suggest that the high expression of miR-410-5p represses the growth of muscle cells. Our results also show that the expression level of miR-410-5p is relatively low, and the expression of *TEAD1* is upregulated, whether in late pregnancy or in the period of rapid muscle development after birth. As discussed earlier, TEAD1 can promote the proliferation of myoblasts, while miR-410-5p can inhibit the proliferation of myoblasts. 

## 5. Conclusions

In conclusion, we constructed a co-expression network using whole transcriptome analysis. Three miRNAs (miR-493-3p, miR-3959-3p, and miR-410-5p), three lncRNAs (MSTRG.3533, MSTRG.4324, and MSTRG.1470), and three genes (*TEAD1*, *ZBTB34*, and *POGLUT1*) were identified as critical components in fibrogenesis in the fetus. Through the functional verification of the DE-miRNAs and target genes at the molecular and cellular levels, it was found that miR-410-5p and the *TEAD1* gene had a significant targeted regulatory relationship. Downregulating the expression of the *TEAD1* gene can significantly inhibit the proliferation of sheep embryonic myoblasts, downregulating the expression of the downstream genes of the *TEAD1* gene and marker genes related to muscle development. The expression levels of the *TEAD1* gene were significantly different in the sheep breeds with different growth rates. These results reveal that the *TEAD1* gene and miR-410-5p could regulate the proliferation of myoblasts and are related with the growth and development of the embryonic muscle of sheep. The results presented here provide new insight into the molecular mechanisms of muscle development.

## Figures and Tables

**Figure 1 cells-12-00943-f001:**
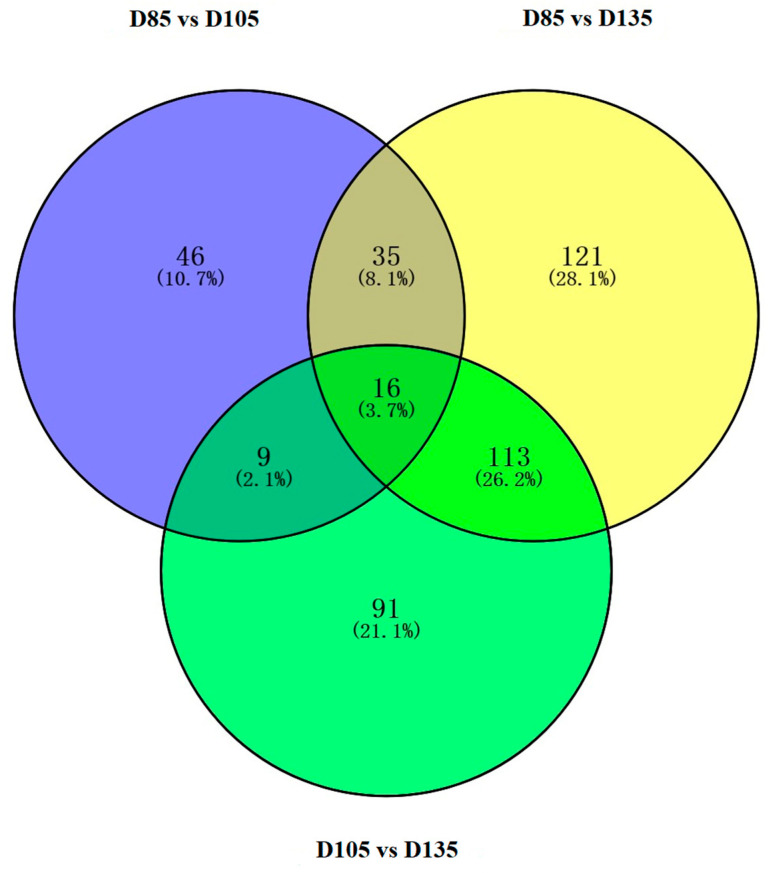
Venn diagrams of the pairwise comparison of differentially expressed miRNA in three groups (D85 vs. D105, D105 vs. D135, and D85 vs. D135).

**Figure 2 cells-12-00943-f002:**
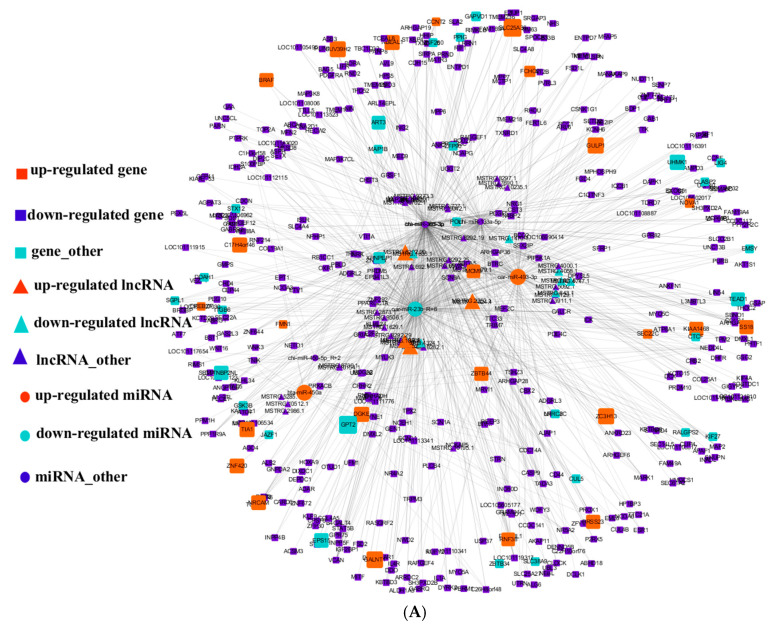
The ceRNA network in the muscle: (**A**) triangles represent lncRNAs, round rectangles represent target genes, and ellipses represent miRNAs, where red represents upregulated expression, green represents downregulated expression, and purple represents neither up- nor downregulated expression; (**B**) a subnet of the ceRNA network.

**Figure 3 cells-12-00943-f003:**
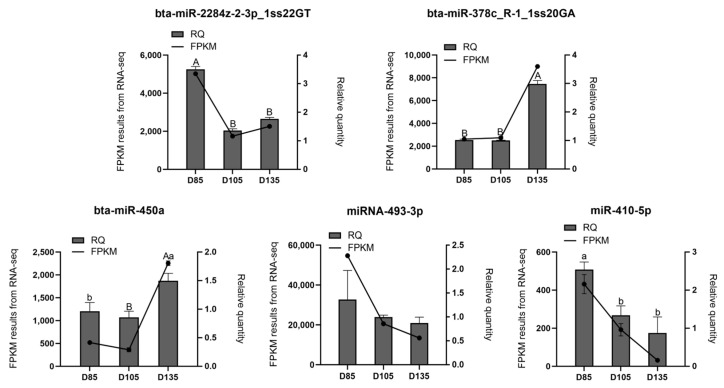
Validation of the small-RNA sequencing data: the polyline graph shows the FPKM value of the sequencing, and the histogram shows the relative quantitative results; a and b indicate a significant difference, and A and B indicate an extremely significant difference.

**Figure 4 cells-12-00943-f004:**
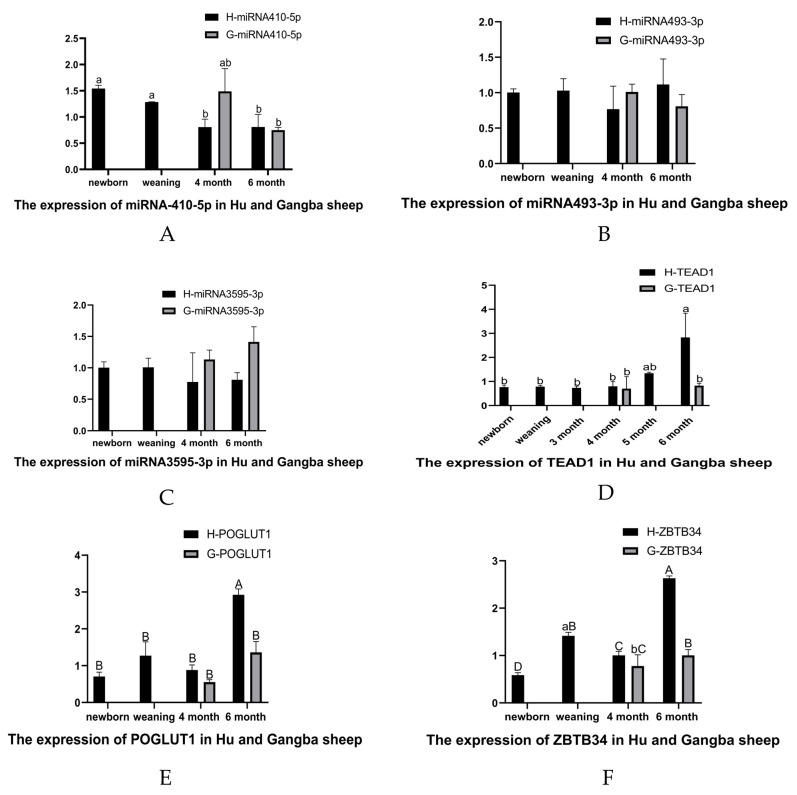
qRT-PCR of DE-miRNAs and target genes in muscle tissues of Hu sheep and Gangba sheep: qRT-PCR results of (**A**) miR-410-5p, (**B**) miR-493-3p, (**C**) miR-3959-3p, (**D**) *TEAD1*, (**E**) *PUGLUT1*, and (**F**) *ZBTB34* in the muscle tissue of Hu sheep and Gangba sheep and the results of the comparative analysis of the muscle tissue expression of the two sheep breeds at 4 and 6 months of age. a and b represent significant results, and A, B, C and D represent extremely significant results.

**Figure 5 cells-12-00943-f005:**
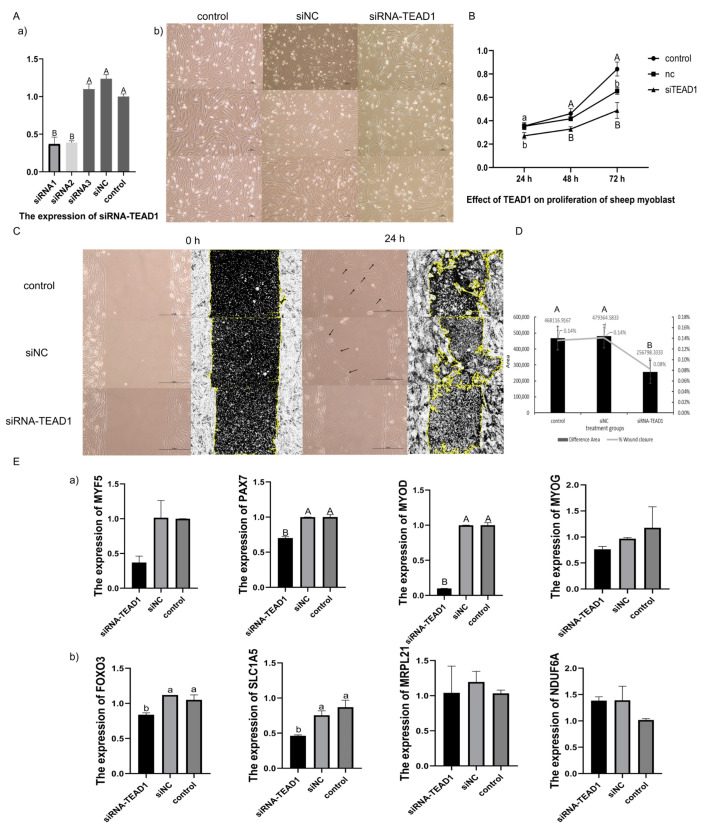
Knocked-down *TEAD1* expression inhibited the proliferation of sheep primary embryonic myoblasts. (**A**) Comparing the transfection effect of three siRNAs and the phenotypes of sheep primary embryonic myoblasts transfected with siRNA1-TEAD1, siNC, and control were analyzed. (**Aa**) the expression of TEAD1 gene. (**Ab**) the phenotypes of sheep primary embryonic myoblasts. Scale bar, 50 μm. (**B**) CCK8 assay of siRNA1–TEAD1 knockdown, where the y-axis is the OD value and the x-axis is time. (**C**) Scratch assay of the siRNA–TEAD1 knockdown. The left picture shows the cell phenotype observed under a microscope, and the yellow highlighted picture on the right shows the use of ImageJ software to analyze the cell damage area. Scale bar, 200 μm. (**D**) The difference (left y-axis) and ratio (right y-axis) in the cell damage area in the different treatment groups of the scratch assay. (**E**) The qRT-PCR results of related genes after siRNA–TEAD1 knockdown: (**Ea**) muscle growth and development-related marker genes—*MYF5*, *Pax7*, *MyoD*, and *MyoG*; (**Eb**) *TEAD1*-related downstream genes—*FoxO3*, *SLC1A5*, *Mrpl21,* and *Ndufa6*. a and b represent significant results, and A and B represent extremely significant results.

**Figure 6 cells-12-00943-f006:**
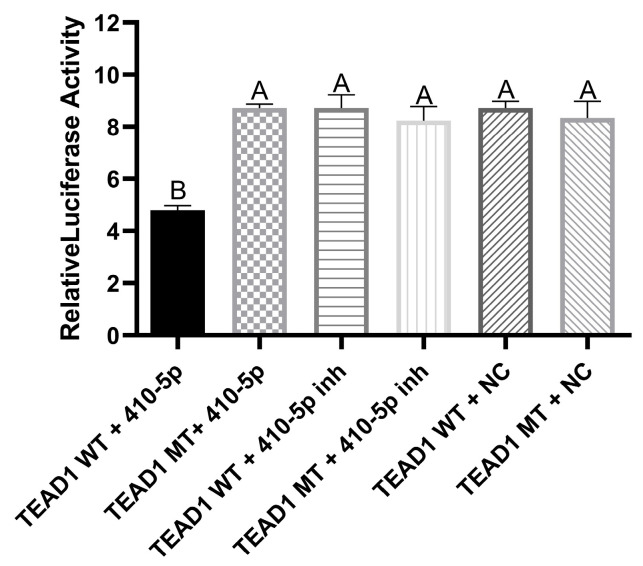
Dual luciferase assay of miR-410-5p and the *TEAD1* gene. A and B represent extremely significant results.

**Figure 7 cells-12-00943-f007:**
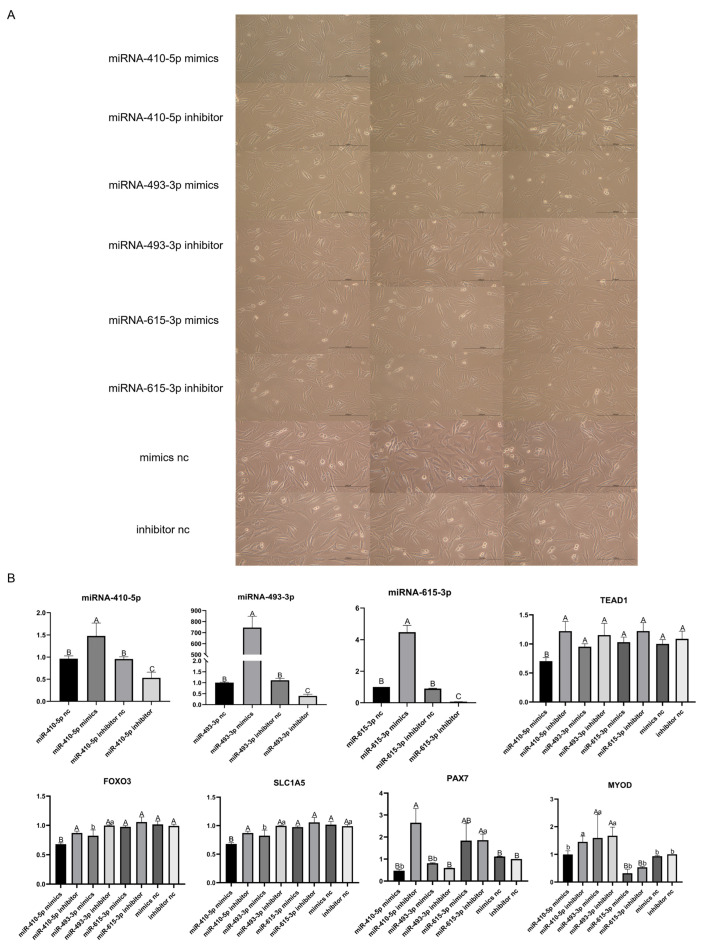
Validation of the targeting relationship between DE-miRNAs and the *TEAD1* gene: (**A**) phenotype pictures of sheep primary embryonic myoblasts transfected with mimics and inhibitor of miRNA410-5p, miRNA493-3p, miRNA615-3p, mimics nc, and inhibitor nc. Scale bar, 200 μm; (**B**) qRT-PCR results of miRNA410-5p, miRNA493-3p, miRNA615-3p, *TEAD1*, *Pax7, MyoD*, *FoxO3,* and *SLC1A5*. a and b represent significant results, and A, B and C represent extremely significant results.

**Figure 8 cells-12-00943-f008:**
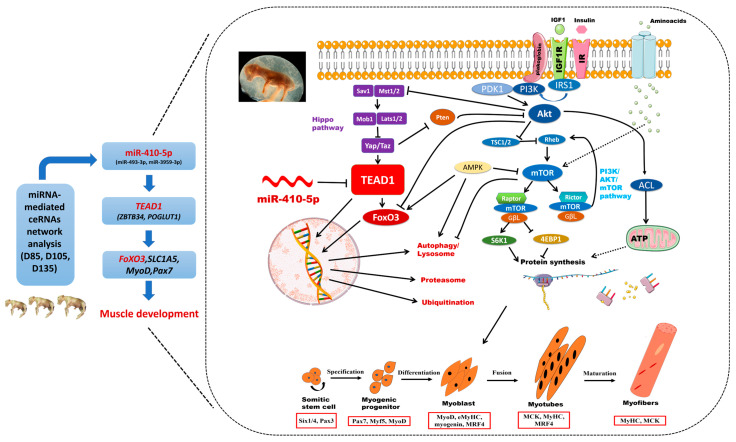
The molecular mechanism of the *TEAD1* gene and miR-410-5p affecting embryonic skeletal muscle development through a miRNA-mediated ceRNA network analysis.

## Data Availability

The raw reads of the RNA samples are available in the NCBI GEO database (accession number: GSE127287). “https://www.ncbi.nlm.nih.gov/geo/query/acc.cgi?acc=GSE127287 (accessed on 18 November 2019)”.

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
