# Peer review of "The Molecular Mechanism of the TEAD1 Gene and miR-410-5p Affect Embryonic Skeletal Muscle Development: A miRNA-Mediated ceRNA Network Analysis"

_cells, 2023, doi:10.3390/cells12060943_

Round 1

Reviewer 1 Report

Overall, this is a clear and well-written manuscript. As we know, key miRNA and genes play important roles in muscle development. This study identified three miRNA and genes affecting muscle development, which provides an important molecular basis for the analysis of the molecular mechanism of sheep muscle development. The methods used in this paper are generally appropriate, and the results are clear. Specific comments are as follows.

1.     p1. line 13: Please check the terminology of double luciferase assay. I think it should be dual luciferase assay? Please also check the vector name.

2.     p4. line 183: “miRNA minics” -> “miRNA mimics”

3.     p10. line 329: Please delete the words “and TEAD1”

4.     p14. line 410: “A and b represent significant results” -> “a and b …..”

5.     p17. line 485: “whose function” -> “which function”

6.     p17. line 504: “grows rapid” -> “grows rapidly”

Author Response

Point 1: line 13: Please check the terminology of double luciferase assay. I think it should be dual luciferase assay? Please also check the vector name.

Response 1: Thank you for your comments, all “double luciferase assay” has been revised to “dual luciferase assay”, and the name of the vector also has been completed in this article.

Point 2: line 183: “miRNA minics” -> “miRNA mimics”

Response 2: Thanks, that was mistake, all “miRNA minics” were changed to miRNA mimics.

Point 3: p10. line 329: Please delete the words “and TEAD1”

Response 3: “and TEAD1” has been deleted, the details were at lines 350-353.

Point 4: p14. line 410: “A and b represent significant results” -> “a and b …..”

Response 4: we have revised this mistake, the details were at lines 442-443.

Point 5: p17. line 485: “whose function” -> “which function”

Response 5: “whose function” has been replaced with“which function”. And all English were polished by a native English speaker.

Point 6: p17. line 504: “grows rapid” -> “grows rapidly”

Response 6: “grows rapid” has been replaced with“grows rapidly” , the details at line 599. Thanks for all your comments.

Reviewer 2 Report

The molecular mechanism of embryonic muscle development is a hot topic in livestock, which is vital for the production and quality of meat. This manuscript focused on sheep muscle development. Based on small RNA-seq data and ceRNA network theory, authors constructed Integral lncRNA-miRNA-mRNA interaction networks and further revealed the regulation of miR-410-5p/TEAD1 in myoblasts proliferation. The data is reliable. The study is based on previous work. Gene mining and validation work were well done. This study will be a good reference for peers who work on muscle development in sheep.
However, some work is still needed, as follows:
1  Please provide detailed information for data analysis, like the qPCR data in Figure 6.
2  Please explain why miR-493-3p inhibitor-treated cells increased the miRNA-493-3P level presented in Figure 9B.
3  There are some errors in the manuscript, such as grammar, punctuation, repetition, unit of measurement, etc. The names of genes and proteins should be unified.
4  what does “N” mean in the description of D85N, D105N, and D135N?
5  Please provide the full names of the relevant genes mentioned in lines 46 to 51 of this paper to facilitate the reader's understanding.

Author Response

Point 1: Please provide detailed information for data analysis, like the qPCR data in Figure 6.

Response 1: Thanks for your suggestion, we revised the material and method, provided detailed information for data analysis,  added “2.1Sequencing and quantitative samples”, and revised “2.7. qRT-PCR of miRNAs and target genes in longissimus dorsi muscle”, provided detailed information of qPCR.

Point 2: Please explain why miR-493-3p inhibitor-treated cells increased the miRNA-493-3P level presented in Figure 9B.

Response 2: Thanks for your suggestion. It was an error, we synthesized a new miRNA-493-3p inhibitor, and we have repeated this experiment in recent days and obtained correct and reliable results, which have been updated in Figure 9. At the same time, here is our raw data. It can be seen that miR-493-3p inhibitor can indeed reduce the expression of miRNA-493-3p in cells.

groups

Reference mean value

Sample mean value

Ct value

Nc reference Ct Mean

ΔΔCt

RQ

RQ mean value

Standard Deviation

mimics

12.498

16.924

4.426

14.152

-9.726

846.8296

746,1446

83.2606848

12.498

17.322

4.823

14.152

-9.329

642.9304

12.498

17.102

4.604

14.152

-9.548

748.674

nc

12.636

26.801

14.165

14.152

0.013

0.9907

1.0004

0.021583213

12.636

26.817

14.181

14.152

0.029

0.980292

12.636

26.745

14.109

14.152

-0.043

1.030385

inhibitor

12.202

27.400

15.198

14.152

1.046

0.484373

0.3979

0.061902745

12.202

27.800

15.598

14.152

1.446

0.367086

12.202

27.900

15.698

14.152

1.546

0.34250367

inhibitor nc

12.127

26.010

13.883

14.152

-0.269

1.204936

1.1120

0.071747329

12.12696

26.236

14.109

14.152

-0.043

1.030223

12.12696

26.13999

14.013

14.152

-0.139

1.101116

Point 3: There are some errors in the manuscript, such as grammar, punctuation, repetition, unit of measurement, etc. The names of genes and proteins should be unified.

Response 3: Those errors have been revised in our manuscript. And we found an English speaker to polish our manuscript.

Point 4: what does “N” mean in the description of D85N, D105N, and D135N?

Response 4: “N” here means all the sheep were on a normal diet, but it is not necessary. In order to avoid misunderstanding, all D85N, D105N, and D135N have been replaced with D85, D105, and D135. Thanks for your suggestion.

Point 5: Please provide the full names of the relevant genes mentioned in lines 46 to 51 of this paper to facilitate the reader's understanding.

Response 5: The full names of the relevant genes were provided, the details at lines 47-51. Thanks for your suggestion.

Reviewer 3 Report

Review of Hu et al
This is an interesting discovery science paper that proposes a competitive endogenous RNA network that helps to regulate muscle development in sheep. Based on this analysis, the investigators knocked down Tead1 by siRNA to investigate its role in muscle development. As a reviewer, I am not competent to judge the strategy for characterising the competitive endogenous RNA network but I can comment on Tead1, Hippo signalling and myogenesis.  

Major comment: Tead1 is part of the Hippo signal transduction network and it is co-regulated by the Hippo effectors Yap and Taz. There is a large amount of literature how Yap/Taz/Tead1-4 regulated the proliferation and differentiation of myoblasts (Sun et al., 2017; Watt et al., 2010), satellite cells (Judson et al., 2012), and muscle mass (Goodman et al., 2015; Watt et al., 2015).

Other comments:
-    Whilst the English is good, the readability is poor primarily because of too many uncommon abbreviations. I recommend always spelling out ceRNA, MRE and SPEM. The readers will enjoy your manuscript more!
-    Avoid overusing words such as “important” or “invaluable” and let the reader come to this conclusion.
-    Figures 2 , 3 and 4 are what some people call “confusograms” with lots of detail but little information for the overall story. Consider adding some of them to the supplemental data. Focus on showing that data that introduce new ideas and that are essential for the story and discovery! See https://www.nature.com/articles/d41586-021-02480-z
-    Figure 10 is useful, but equally misleading. Tead is primarly co-activated by the Hippo effectors Yap and Taz which is not shown in the figure.
References
Goodman, C.A., Dietz, J.M., Jacobs, B.L., McNally, R.M., You, J.-S., and Hornberger, T.A. (2015). Yes-Associated Protein is up-regulated by mechanical overload and is sufficient to induce skeletal muscle hypertrophy. FEBS letters 589, 1491-1497.
Judson, R.N., Tremblay, A.M., Knopp, P., White, R.B., Urcia, R., De Bari, C., Zammit, P.S., Camargo, F.D., and Wackerhage, H. (2012). The Hippo pathway member Yap plays a key role in influencing fate decisions in muscle satellite cells. Journal of cell science 125, 6009-6019. 10.1242/jcs.109546.
Sun, C., De Mello, V., Mohamed, A., Ortuste Quiroga, H.P., Garcia-Munoz, A., Al Bloshi, A., Tremblay, A.M., von Kriegsheim, A., Collie-Duguid, E., Vargesson, N., et al. (2017). Common and Distinctive Functions of the Hippo Effectors Taz and Yap in Skeletal Muscle Stem Cell Function. Stem cells 35, 1958-1972. 10.1002/stem.2652.
Watt, K.I., Judson, R., Medlow, P., Reid, K., Kurth, T.B., Burniston, J.G., Ratkevicius, A., De Bari, C., and Wackerhage, H. (2010). Yap is a novel regulator of C2C12 myogenesis. Biochemical and biophysical research communications 393, 619-624. 10.1016/j.bbrc.2010.02.034.
Watt, K.I., Turner, B.J., Hagg, A., Zhang, X., Davey, J.R., Qian, H., Beyer, C., Winbanks, C.E., Harvey, K.F., and Gregorevic, P. (2015). The Hippo pathway effector YAP is a critical regulator of skeletal muscle fibre size. Nature communications 6, 6048.

Author Response

Major comment:

Point 1: Tead1 is part of the Hippo signal transduction network and it is co-regulated by the Hippo effectors Yap and Taz. There is a large amount of literature how Yap/Taz/Tead1-4 regulated the proliferation and differentiation of myoblasts (Sun et al., 2017; Watt et al., 2010), satellite cells (Judson et al., 2012), and muscle mass (Goodman et al., 2015; Watt et al., 2015).

Response 1: Thank you very much for your suggestion, this is a particularly great suggestion. We have consulted a large number of literature about the Hippo signal transmission network, and have carried out a detailed review and discussion in the discussion part of this article. The relevant literature you have provided has been cited, and we have also cited much literature that we think is very valuable. We also redrew Figure 10 and added the key protein of the hippo pathway, which is indeed more clear and more complete at present. Thanks again.

Point 2: Whilst the English is good, the readability is poor primarily because of too many uncommon abbreviations. I recommend always spelling out ceRNA, MRE and SPEM. The readers will enjoy your manuscript more!

Response 2: The full name of ceRNA, MRE, and sheep primary embryonic myoblasts (SPEM) was provided in this article, in addition, we have revised the article, changing the uncommon abbreviations into full names for easy reading and understanding, the details at lines 19, 68 and 73. Thank you for your suggestions.

Point 3: Avoid overusing words such as “important” or “invaluable” and let the reader come to this conclusion.

Response 3: Thanks for your suggestion, we have noticed this problem and revised the full text to avoid using such words as far as possible.

Point 4: Figures 2, 3 and 4 are what some people call “confusograms” with lots of detail but little information for the overall story. Consider adding some of them to the supplemental data. Focus on showing that data that introduce new ideas and that are essential for the story and discovery! See https://www.nature.com/articles/d41586-021-02480-z

Response 4: Your opinion is very correct. It is indeed true. The figure looks very complex and the information directly extracted is less, we've added figures 2 and 3 to the supplemental figures, and they named Figure S2 and Figure S4, respectively.

Point 5: Figure 10 is useful, but equally misleading. Tead is primarly co-activated by the Hippo effectors Yap and Taz which is not shown in the figure.

Response 5: Thanks for your suggestion, we improved Figure 10 and added the primary co-activated effectors Yap and Taz, and also add the hippo pathway, which is indeed more clear and more complete at present.

References

Goodman, C.A., Dietz, J.M., Jacobs, B.L., McNally, R.M., You, J.-S., and Hornberger, T.A. (2015). Yes-Associated Protein is up-regulated by mechanical overload and is sufficient to induce skeletal muscle hypertrophy. FEBS letters 589, 1491-1497.
Judson, R.N., Tremblay, A.M., Knopp, P., White, R.B., Urcia, R., De Bari, C., Zammit, P.S., Camargo, F.D., and Wackerhage, H. (2012). The Hippo pathway member Yap plays a key role in influencing fate decisions in muscle satellite cells. Journal of cell science 125, 6009-6019. 10.1242/jcs.109546.
Sun, C., De Mello, V., Mohamed, A., Ortuste Quiroga, H.P., Garcia-Munoz, A., Al Bloshi, A., Tremblay, A.M., von Kriegsheim, A., Collie-Duguid, E., Vargesson, N., et al. (2017). Common and Distinctive Functions of the Hippo Effectors Taz and Yap in Skeletal Muscle Stem Cell Function. Stem cells 35, 1958-1972. 10.1002/stem.2652.
Watt, K.I., Judson, R., Medlow, P., Reid, K., Kurth, T.B., Burniston, J.G., Ratkevicius, A., De Bari, C., and Wackerhage, H. (2010). Yap is a novel regulator of C2C12 myogenesis. Biochemical and biophysical research communications 393, 619-624. 10.1016/j.bbrc.2010.02.034.
Watt, K.I., Turner, B.J., Hagg, A., Zhang, X., Davey, J.R., Qian, H., Beyer, C., Winbanks, C.E., Harvey, K.F., and Gregorevic, P. (2015). The Hippo pathway effector YAP is a critical regulator of skeletal muscle fibre size. Nature communications 6, 6048.

Author Response

My major concerns are:

Point 1:The missing information of sheep primary embryonic myoblasts. Are they purified myoblasts (cloned, fluorescence activated cell sorted myoblasts)? Are they characterized (i.e differentiated to myotubes with multiple nuclei)? Without detailed information of sheep primary embryonic myoblasts, it is very difficult to interpret the results because the data might come out from other types of cells in culture. Thus, generation of sheep primary embryonic myoblasts, passage information and differentiating information may be needed for publication.

Response 1: Thanks for your great suggestion. It is true that our manuscript does not introduce this cell line in detail. The cell line used in this study was sheep primary embryonic myoblasts that were cultured, purified, and characterized from sheep embryos. We purified myoblasts and characterized this cell lines, it could be differentiated to myotubes with multiple nuclei. The relevant research results were not only published in the Journal of Genomics and Applied Biology [1], but also granted two related patents[2-3]. We have revised the article, added the literature, and introduced the cell line in the material method.

  1. Liu, Lulu; Hu, Wenping; Shang, Mingyu; Wang, Xinyue; Feng, Mian; Liu, Yunhui; Xiong, Jinko; Zhang, Li. Improvement and characterization of embryo-derived myogenic cells from lake sheep in culture. Genomics and Applied Biology 2022, 41, 1440-1451, doi:10.13417/j.gab.041.001440.
  2. Zhang Li; Hu Wenping; Liu Lulu; Shang Mingyu; Wang Xinyue; Feng Mian; Xiong Jinko; A high-efficiency method for primary cell culture of sheep embryonic skeletal muscle, 2021-7-1, China, ZL202110740406.3 (Chinese National Invention Patent).
  3. Zhang Li; Hu Wenping. A method of transmission and purification of myoblasts derived from sheep embryos.2022-10-4, China, ZL202210506931.3(Chinese National Invention Patent).

Point 2:The results from sheep primary embryonic myoblasts may not completely reflect the reality of embryonic muscle development. Thus, authors may need to adjust the statements, or present the limitations of the study.

Response 2: Thanks for your suggestion, We have modified the related statement, the details were at lines 23-25 and 634-638.

Point 3:Figure 5, please present the validation RNA-seq data of miR-410-5p, and TEAD1 gene here if authors have done. If not, please explain it.

Response 3: Thanks for your suggestion. We have added the miRNA-410-5p quantitative validation results in Figure 5. In this paper, small RNA sequencing was performed, so figure 5 showed the Validation of Small-RNA sequencing data, so here we did not put the TEAD1 result in this manuscript. We are writing another manuscript about the analysis of the mRNA sequencing data, we will put the related results in that paper. Here is the TEAD1 result in embryo, it is reversed with miR-410-5p too(figure 1 ). From 85 to 135 days of embryonic development, the expression of miR410-5p gradually decreased, while the expression of TEAD1 gradually increased.

Figure 1. The expression of TEAD1 in Chinese Merino embryo

Point 4:An extended discussion of miR-410-5p and miRNAs may help understand the interactions of miRNAs and their target genes.

Response 4: Thanks for your suggestion. We revised the discussion. The research about miR-410-5p is very few, but it does related to muscle cells, such as the inhibition of miR-410-5p decreased cardiac fibrosis. Although there are few studies on the functions of miR-410-5p, this also shows that this paper has certain reference significance. We have added the discussion to help understand the interactions of miRNAs and their target genes.

Minors:

Point 1: Figure 6, it seems like that the central column data are a duplicate of part from the right column data though the central column data might be processed independently. It may cause confusion for readers.

Response 1: Thank you very much for your comments, we have revised and improved Figure 6. Now, the revision version of Figure 6 is Figure 4.

Point 2: Extra TEAD1 in line 329?

Response 2: The Extra TEAD1 gene was deleted, the details at 420-423.

Point 3: Duplications in line 347?

Response 3: This mistake was revised, the details at 370-376.

Thanks for all your suggestion.

Author Response

Point 1: In the Materials and methods chapter, exact name and type of skeletal muscle is only mentioned in the first section (under smallRNA sequencing). However, there is no information about the skeletal muscles (name, type, age, species) in the following sections, so it is not clear whether the same muscles were used for further analysis or not. I suggest to create a separate section about the detailed description of muscle types and ages in the Materials and methods.

Response 1: Thank you for your valuable comments, we have provided a detailed description of the tissue samples used for sequencing and validation in the Materials Methods section, the details at 87-99.

Point 2: What was the reason for using 3 different species for qPCR experiments? Which muscle was used for qPCR analysis?

Response 2: It is a very good question. Chinese Merino, Hu and Gangba sheep are are very representative sheep breed. Chinese Merino sheep is from Xingjiang province, Northwest of China, are large and fast growing. It is a cultivar, the main characteristics are their large body size and high amount of wool. Hu sheep are medium-sized and grow fast in the first 6 months, Hu sheep is a local breed in the the Taihu Lake(Southeast of China). Ganba sheep are from Tibet (Southwest of China), and are small and slow-growing, but Ganba sheep adapting to high altitude environment well. They are excellent materials for conducting comparative studies on muscle growth and development:

Chinese Merino sheep are large in size: Body weight, adult rams average 113.15 kg, ewes 56.45 kg;

Hu sheep are medium-sized with fast early growth and development: adult rams weigh 79.3kg, ewes 50.6kg, 5-6-month-old lambs 45kg for rams and 36kg for ewes;

Gangba sheep grows slow and the weight of at the age of 24 and 36 months are 25 and 35 kg respectively.

The longissimus dorsi was used for all qRT-PCR validation, we have added the information in this article.

Point 3: Why has only one type of muscle been used? M. Longissimus dorsi is composed of type I fibers, but there is no information about type II fibers.

Response 3: Thanks for your suggestion. Primary myofibers form during the initial stage of myogenesis in embryonic development. Secondary myofibers form during the second wave of myogenesis in the fetal stage and account for the majority of skeletal muscle fibers. Postnatal muscle growth is mainly due to an increase in muscle fiber size without forming new muscle fibers[1]. Myofibre types switch during the development of skeletal muscle. The change in muscle fiber type is a more complicated problem, so here we only used Longissimus dorsi as our experiment materials, to make it simple. But The types of muscle fibers will be further studied in our future experiments.

       Reference

  1. Du M, Tong J, Zhao J, Underwood KR, Zhu M, Ford SP, Nathanielsz PW. Fetal programming of skeletal muscle development in ruminant animals. J Anim Sci. 2010 Apr;88(13 Suppl):E51-60. doi: 10.2527/jas.2009-2311. Epub 2009 Aug 28. PMID: 19717774.

Point 4: Quantitative real-time PCR sometimes is abbreviated as QRTPCR, while in other cases as RT-qPCR. I suggest to use the same acronym throughout the text, unless they used two slightly different versions of qPCR (quantitative real-time PCR versus reverse transcription quantitative real-time PCR).

Response 4: Thanks for your suggestion. We have uniformed the name of quantitative real-time PCR to qRT-PCR.

Point 5: Line 212 has an incorrect composition: „over half clean reads (52,6%)”.

Response 5: We have modified this mistake, thanks. The details was at line 243.

Point 6: Figure/Table is missing about miRNA expression profile in section 3.1 of the Results chapter, therefore no information is available about the 2,275 newly detected miRNAs.

Response 6: The raw data of the miRNA expression profile and the 2,275 newly miRNAs were provided in Table S3.

Point 7: Also, there is no information about DE genes (which DE genes belong to the different types) (section 3.2, Line 218-222).

Response 7: Thanks for your suggestion. All data of the DE gene have been uploaded to the database(Transcriptomic accession:GSE127287).

 https://www.ncbi.nlm.nih.gov/geo/query/acc.cgi?acc=GSE127287 

Here we concentrated on the analysis of differential miRNAs in this article, but we used data from DEgene to help analyze the ceRNA network. Our other article will provide a very detailed analysis of the DE gene, which is not presented in this article. And The raw data of the DE miRNAs were provided in Table S4, all the details were at lines 246-250.

Point 8 : Which internal control was used in the RT-qPCR for the validation of RNA-Seq data? (Results 3.6) Are the results normalised to U6 or β-actin level? Is there any proof that the levels of internal controls do not change during muscle development?

Response 8: U6 was used in qRT-PCR for the validation of RNA-Seq data, the levels of U6 controls do not change during muscle development, and the raw data of U6 are as follows:

Developmental stage

GD85

GD105

GD135

Quantitative average

13.374

12.378

12.841

Point 9 : Is there any evidence that TEAD1 gene and miR-410-5p has any function in postnatal skeletal muscle development?

Response 9: No studies have shown a role for miRNA-410-5p and TEAD1 genes in postnatal sheep skeletal muscle. This is the first study about the miRNA-410-5p and TEAD1 genes in postnatal sheep skeletal muscle, the qRT-PCR analysis results of Figure 6 showed that The expression trend of miR-410-5p and TEAD1 gene is reversed, and the TEAD1 gene showed a significant correlation with postnatal skeletal muscle growth and development in sheep. These results provide preliminary evidence that the TEAD1 gene and miR-410-5p may have any function in postnatal skeletal muscle development. We added many discussion about TEAD1 and miR-410-5p in the discussion part. Thanks for all your suggestion.

Round 2

Reviewer 1 Report

Agree to publish.

Reviewer 2 Report

No more comments.

Reviewer 4 Report

The authors addressed my concerns, and the manuscript revised accordingly.

I have no further comments.

Reviewer 5 Report

Thank you for your responses, I accept the revised paper.